# ACE and UCP2 gene polymorphisms and their association with baseline and exercise-related changes in the functional performance of older adults

Justin W.L. Keogh[1,2,3], Barry R. Palmer[4,5], Denise Taylor[6] and Andrew E. Kilding[2,7]

[1] Faculty of Health Sciences and Medicine, Bond University, Australia
[2] Human Potential Centre, AUT University, Auckland, New Zealand
[3] Faculty of Science, Health, Education and Engineering, University of the Sunshine Coast, Australia
[4] Christchurch Heart Institute, Department of Medicine, University of Otago, Christchurch, New Zealand
[5] Institute of Food, Nutrition and Human Health, College of Health, Massey University Wellington, New Zealand
[6] Health and Rehabilitation Research Institute, AUT University, Auckland, New Zealand
[7] School of Sport and Recreation, AUT University, Auckland, New Zealand

Corresponding author
Justin W.L. Keogh,
jkeogh@bond.edu.au

## ABSTRACT

Maintaining high levels of physical function is an important aspect of successful ageing. While muscle mass and strength contribute to functional performance in older adults, little is known about the possible genetic basis for the heterogeneity of physical function in older adults and in how older adults respond to exercise. Two genes that have possible roles in determining levels of muscle mass, strength and function in young and older adults are angiotensin-converting enzyme (ACE) and mitochondrial uncoupling protein 2 (UCP2). This study examined whether polymorphisms in these two individual genes were associated with baseline functional performance levels and/or the training-related changes following exercise in previously untrained older adults. Five-eight Caucasian older adults (mean age 69.8 years) with no recent history of resistance training enrolled in a 12 week program of resistance, balance and cardiovascular exercises aimed at improving functional performance. Performance in 6 functional tasks was recorded at baseline and after 12 weeks. Genomic DNA was assayed for the ACE intron 16 insertion/deletion (I/D) and the UCP2 G-866A polymorphism. Baseline differences among genotype groups were tested using analysis of variance. Genotype differences in absolute and relative changes in physical function among the exercisers were tested using a general linear model, adjusting for age and gender. The genotype frequencies for each of the studied polymorphisms conformed to the Hardy-Weinberg equilibrium. The ACE I/D genotype was significantly associated with mean baseline measures of handgrip strength (II $30.9 \pm 3.01$ v. ID $31.7 \pm 1.48$ v. DD $29.3 \pm 2.18$ kg, $p < 0.001$), 8ft Up and Go time (II $6.45 \pm 0.48$ v. ID/DD $4.41 \pm 0.19$ s, $p < 0.001$) and 6 min walk distance (II $458 \pm 28.7$ v. ID/DD $546 \pm 12.1$m, $p = 0.008$). The UCP2 G-866A genotype was also associated with baseline 8ft Up and Go time (GG $5.45 \pm 0.35$ v. GA $4.47 \pm 0.26$ v. AA $3.89 \pm 0.71$ s, $p = 0.045$). After 12 weeks

of training, a significant difference between UCP2 G-886A genotype groups for change in 8ft Up and Go time was detected (GG $-0.68 \pm 0.17$ v. GA $-0.10 \pm 0.14$ v. AA $+0.05 \pm 0.31$ s, $p = 0.023$). While several interesting and possibly consistent associations with older adults' baseline functional performance were found for the ACE and UCP2 polymorphisms, we found no strong evidence of genetic associations with exercise responses in this study. The relative equivalence of some of these training-response findings to the literature may have reflected the current study's focus on physical function rather than just strength, the relatively high levels of baseline function for some genotype groups as well as the greater statistical power for detecting baseline differences than the training-related changes.

## INTRODUCTION

Maintaining adequate levels of muscle mass, strength, muscular and aerobic endurance and functional performance in older age is important, as a decline in these physical attributes may result in: (1) a loss of independence (*Baumgartner et al., 2004*; *Kim et al., 2012*), (2) an increased risk and fear of falls (*Brouwer, Musselman & Culham, 2004*; *Wagenaar, Keogh & Taylor, 2012*); (3) an increase in the risk of chronic conditions (*Abellan van Kan et al., 2009*); and (4) a reduction in quality of life (*Giles, Hawthorne & Crotty, 2009*; *Masel et al., 2009*). Engaging in regular resistance and aerobic activity and ensuring adequate nutritional intake appear to be some of the key strategies for older adults to reduce these sarcopenic-related losses of muscle mass, strength and function (*Fiatarone-Singh, 2002*; *Nelson et al., 2007*). While initial studies in this area focused on maintaining muscle mass, a number of recent reviews indicate older adults at risk of physical decline may gain more substantial improvements in muscular strength, balance and functional performance, for example gait speed, than improve their muscle mass (*Keogh & MacLeod, 2012*; *Valenzuela, 2012*). This is vital as the age-related loss of strength, balance and gait speed has a greater relationship to outcomes such as activities of daily living, independence and quality of life and mortality than muscle mass alone (*Abellan van Kan et al., 2009*; *Kim et al., 2012*; *Wood et al., 2011*).

Studies involving younger (*Argus et al., 2009*; *Till & Cooke, 2009*) and middle-aged to older (*Karavirta et al., 2011*) adults indicate the potential for considerable inter-individual responses to identical training programs, even when the sample is relatively homogenous at baseline and engage in the same exercise program. For example, in a study in which 175 untrained middle-aged to older adults performed 21 weeks of strength, aerobic training, combined strength and aerobic or no training, *Karavirta et al. (2011)* reported large individual differences in muscular strength ($-12$ to $87\%$) and aerobic power ($-8$ to $42\%$) changes for the combined strength and endurance training group ($n = 53$).

This high degree of within-participant variability in response was demonstrated via the use of criterion assessments of strength and aerobic power, namely the maximum voluntary contraction (MVC) and cycle ergometry peak oxygen consumption (VO$_{2peak}$) tests, respectively. Further, there were no significant correlations (all r's $< 0.10$) between the changes in the MVC strength or VO$_{2peak}$ in any of the three groups, suggesting that the strength and aerobic power responses to strength and/or aerobic training can exhibit substantial within- and between-participant variability. These intra-individual differences in training response may reflect genetic factors.

While considerable heterogeneity exists in the physical function of many older adults, even when chronic conditions and medications are controlled for, the relative contribution of genetics and long-term physical activity levels in determining their physical function and ultimately survival remains less clear. Based on research involving primarily younger adults, two genes that might play a role in regulating the physical function and indirectly impact on survival in older adults are angiotensin-converting enzyme (*ACE*) and mitochondrial uncoupling protein 2 (*UCP*2) (*Dhamrait et al., 2012*; *Puthucheary et al., 2011*).

When expressed, the *ACE* gene produces angiotensin-converting enzyme protein. This *ACE* protein plays a key role in regulating the activity of the rennin-angiotensin system, thereby directly influencing blood pressure and fluid balance and indirectly influencing cardiovascular and musculoskeletal structure and function (*Puthucheary et al., 2011*; *Seripa et al., 2011*). Indeed, a functional polymorphism in the human *ACE* gene, the intron 16 insertion (I) allele, has been associated with an enhanced cardiovascular response to training (*Myerson et al., 1999*), while the deletion (D) allele has been associated with a superior muscle size and strength response to training (*Puthucheary et al., 2011*; *Woods et al., 2001*).

The *UCP2* gene is expressed in many tissues particularly skeletal muscle. It exerts a variety of effects on mitochondrial function, meaning it influences the rate of synthesis of ATP and reactive oxygen species and indirectly influences many indices of cardiovascular health (*Bo et al., 2013*; *Dato et al., 2014*; *Dhamrait et al., 2004*; *Palmer et al., 2003*). The frequency of the *UCP2* (A55V) C allele was found to be higher among power athletes compared with controls (*Sessa et al., 2011*). Another allele of the *UCP2* G-866A polymorphism has also been associated with increased delta efficiency after endurance training in young, healthy adults compared to those carrying the G allele (*Dhamrait et al., 2012*; *Perusse et al., 2013*).

Recently, some studies have examined the relationships between *ACE* (*Bustamante-Ara et al., 2010*; *Garatachea et al., 2012*; *Giaccaglia et al., 2008*; *Pereira et al., 2013*) and *UCP2* (*Dato et al., 2014*) gene polymorphisms and a variety of measures of muscular strength and power in older adults. Only one of these four *ACE* studies reported significant genotypic relationship to performance at baseline, with the *ACE* insertion/deletion (ACE ID) group having a significantly greater baseline handgrip strength (*Bustamante-Ara et al., 2010*). In support of a possible genetic link to baseline function, *Seripa et al. (2011)* demonstrated that the *ACE* II genotype was associated with increased risk of limitations in activities of daily living in hospitalised older patients. Of the two studies that assessed training-related changes in older adults, both reported significantly greater improvements

in strength and power measures for those with the *ACE* D allele (*Giaccaglia et al., 2008*; *Pereira et al., 2013*). To the authors' knowledge, only one study has so far reported associations between baseline differences in older adults who differ in *UCP*2 genotypes (*Dato et al., 2014*), with no studies examining the exercise response. *Dato et al. (2014)* found significantly greater walking speeds over 3–4 m in those with the rs7109266 SNP found on the *UCP2* G-866A allele, but no *UCP2* association with the degree of disability, handgrip strength or cognitive function in a sample of 1089 older adults.

Therefore, the purpose of this study was to determine if either the *ACE* and/or *UCP2* genotypes individually: (1) distinguish the baseline level of functional performance in older adults who are not currently performing an exercise program; and (2) influence the training-related changes in older adults' functional performance. It is hypothesised that variations in these individual alleles will be associated with baseline- and training-related differences in older adults' functional performance.

## MATERIALS AND METHODS

### Research design

This study utilised a cross-sectional and pre-post single arm trial design to determine whether polymorphisms in two individual genes (*ACE* and *UCP2*) were associated with baseline and training-related changes in a variety of physical performance measures in community dwelling older adults, respectively. Due to a lack of sufficient statistical power, no interactions between polymorphisms of the two genes and the physical performance measures were examined. Portions of this data describing the exercise program, physical assessments used and the magnitude of changes in physical function seen with training have been reported previously (*Keogh et al., 2014*).

### Participants

As the Never Too Old (N2O) program is a community-based program for older adults, there was no specific inclusion/exclusion criteria for participation in this study, besides being at least 60 years of age and healthy enough to be given medical clearance to participate in the program. Fifty-eight older adults of European ancestry who had just enrolled in a N2O program gave written informed consent to participate in this study. These 58 participants were a subset of the 67 participants who gave informed consent to participate in Study 1 (which examined the training-related changes in physical function) of the wider N2O study (*Keogh et al., 2014*). The nine participants whose data is not included in this study either declined to participate in the current genetic project or were ineligible to participate based on their ethnicity.

Prior to starting the N2O program, all subjects completed a modified PAR-Q pre-exercise health assessment questionnaire to determine if the potential participants had any relative or absolute contraindications to exercise. If the PAR-Q raised any concerns about the safety of exercise, the individual was required to obtain clearance from a qualified medical practitioner prior to entry into the program. Ethical clearance for the project was approved by the Auckland University of Technology Human Ethics Committee (06/05).

## TRAINING PROGRAM

Participants attended the N2O program twice weekly for 12 weeks, with each training session lasting approximately 60 min. The participants in this research project performed the Bronze N20 program, which focused on providing a safe and friendly environment that introduced the older adults to selected resistance and cardiovascular exercises that were aimed at improving their overall functional performance and health.

Training sessions commenced with a 5–10 min warm-up that preceded the resistance training component. The resistance training exercises included the knee extension, leg curl, leg press, chest press, lat pulldown, shoulder press, bicep curl and tricep pushdown, with a selection of these exercises performed during each training session. Each exercise was performed for 1–2 sets of 8–12 repetitions with loads that initially produced moderately light to moderate ratings of perceived on the Borg rating of perceived exertion (RPE) scale (*Borg, 1982*). After completing the resistance training exercises, 5–10 min of cardiovascular exercise (i.e., stationary cycling or walking on a treadmill) was then performed at a moderately light to moderate intensity on the Borg RPE scale (*Borg, 1982*). The exercise session was then completed by performing 5–10 min of stretches for the major muscle groups of the body. Loads for the resistance and cardiovascular exercises were progressively increased over the course of the 12 week training program.

### Procedures

The N2O program is aligned with the International Society of Aging and Physical Activity (ISAPA) and adopts the ISAPA's recommended Senior Fitness Test battery (*Rikli & Jones, 1999a*; *Rikli & Jones, 1999b*; *Rikli & Jones, 2001*). This series of assessments has been to shown to be reliable and valid in predicting functional levels in older adults and involves the: (1) 30 s sit to stand (30 s STS); (2) 30 s bicep curl; (3) 8 ft Up and Go; and (4) 6 min walk tests. The 30 s STS and bicep curl stand tests were designed to assess lower and upper limb muscular strength and endurance, respectively, of older adults (*Rikli & Jones, 1999a*; *Rikli & Jones, 1999b*). The 8 ft Up and Go and the 6 min walk tests assess dynamic balance/mobility and walking endurance, respectively (*Rikli & Jones, 1999a*; *Rikli & Jones, 1999b*). Measures of upper limb dexterity (Purdue Pegboard Test) and strength (handgrip strength) were also assessed, as older adults have reduced upper limb dexterity and strength compared to young adults (*Keogh, Morrison & Barrett, 2007*; *Sequeria, Keogh & Kavanagh, 2012*). In addition, the height and body mass of each participant was also obtained using standard procedures so that a body mass index (BMI) could be calculated. BMI was calculated by dividing the body mass in kg by the square of the height in metres.

All of these tests were conducted within the same assessment session that were completed in <1 h in the same order as described above with the exception that the 6 min walk test was performed last due to its fatiguing nature. Prior to performing these eight assessments, all participants completed a 5–10 min warm-up that consisted of general total body movements and stretches of primary muscle groups. A description of the eight tests is provided below. Prior to performing each of the tests, participants were given the same instructions, these being "to do the best they can on the tests, but to never

push themselves to the point of overexertion or beyond what they think is safe for them" (*Rikli & Jones, 1999a*).

### 30 s STS

The participant was asked to fully stand and sit down as many times as possible in 30 s from a chair without arms, a seat height of 43 cm and with their arms crossed at the wrist and held against the chest. One practice trial, of 2–3 repetitions, was performed as a specific warm-up followed by one full trial of this test. The score was the complete number of correct repetitions performed in 30 s.

### 30 s Bicep curl

The participant was asked to sit down on a chair and to complete as many bicep curls in 30 s as possible with their preferred arm. This test was done with 2 kg or 4 kg dumbbells for older women and men, respectively. The participant was required to keep their trunk still and to move their forearm through a full range of motion about the elbow joint for each repetition to be counted. One practice trial of 2–3 repetitions (if possible) were performed as a specific warm-up followed by one full trial of this test. The score was the complete number of correct repetitions performed in 30 s.

### 8 ft Up and Go

The 8 ft Up and Go required the participant who was initially seated on a chair that was ∼43 cm high and no arm rests, to stand up, walk around a cone positioned 8 ft (∼2.44 m) away and sit down again in the chair. One practice trial was performed as a specific warm-up followed by two trial of this test, each separated by a rest period of 60 s. The time required to complete each trial of this test was measured via stopwatch to the nearest 0.01 s.

### 6 Min walk

The 6 min walk test was performed by having several older adults starting at different points of a 50 m rectangular course. This course was marked with cones every 5 m to facilitate easy recording of distance walked. One trial was performed due to the challenging and fatiguing nature of this test.

### Purdue Pegboard Test: (PPT)

The upper limb dexterity of the older adults was assessed by the PPT (*Tiffin & Asher, 1948*). The PPT is a time-based dexterity assessment tool comprising four sub-tasks; right hand only, left hand only, both hands and an assembly task, in which the participant is required to place the maximum number of pegs in the holes on the pegboard in a set time. For the handed tests, the participant was required to place pegs into evenly spaced holes on the board using the indicated hand. For the assembly task there is an additional component with the assembly made up of the pegs, washers and collars, placed in a specific order. The individual and both handed tasks are performed for 30 s while the assembly task allows 60 s due to its greater complexity. Each of the sub-tasks was performed 2 times to reduce possible learning effect and improve the reliability of the test (*Tiffin & Asher, 1948*).

### Isometric handgrip strength

Isometric MVC grip strength of the preferred limb was assessed with a hand-grip dynamometer (*Bohannon, 2002*). Participants performed one practice and two test trials of the isometric hand-grip strength test. For all trials, the preferred arm had to be kept by the side of the participant with the elbow maintained at ~90° of elbow flexion. A rest period of 60 s was given between each trial.

## DNA extraction and genotyping

For genetic analysis buccal cells were harvested from participants via a mouthwash of 10 ml 4% sucrose in sterile deionised water. The cell suspension was recovered in a 50 ml tube, vortexed vigorously and stored at −20°C until DNA isolation was performed as previously reported (*Palmer & Healy, 1993*).

DNA samples (100 ng) were genotyped for the *ACE* intron 16 insertion/deletion (I/D) polymorphism using a two primer polymerase chain reaction (PCR) protocol (*Rigat et al., 1992*) with the addition of 5% DMSO to eliminate mis-amplification of heterozygous templates (*Fogarty et al., 1994*). A PCR-RFLP assay (*Esterbauer et al., 2001*) was employed to genotype 100 ng DNA samples for the *UCP*2 G-866A polymorphism, in which 360 bp amplimers were digested with *Mlu*I. Amplimers were electrophoresed on 1 or 2% agarose, 0.5 x TBE gels stained with ethidium bromide and visualized with a Bio-Rad Fluor-S imaging system (Bio-Rad, Hercules, California, USA). A random subset (20%) of the samples were regenotyped in separate assays with 100% concordance.

## Statistical analysis

Data are presented as mean ± standard deviation (SD). Data were analysed using a statistical software package (SPSS, v14, Chicago, Illinois, USA). Baseline differences among genotype groups were tested using analysis of variance. Genotype differences in absolute and relative changes in physical function among the exercisers were tested using a general linear model, adjusting for age and gender. Additive and D allele-dominant (*ACE*) and additive and A allele-dominant (*UCP2*) genetic models were all evaluated. Statistical significance was set at $p < 0.05$. As calculated using G*Power 3.1.3 (*Faul et al., 2009*), the study had greater than 90% power to detect a difference in (for example) baseline Up & Go performance between the *ACE* DD/ID and II genotype groups with an $\alpha = 0.05$. However, the study only had 70% power to detect a difference in post-training 8 ft Up & Go performance between these groups with an $\alpha = 0.05$.

## RESULTS

A description of the participants' baseline demographic data is provided in Table 1. Almost all participants had at least one chronic disease, with high blood pressure (19%), arthritis and other joint conditions (16%) and cardiovascular disease (12%) being most common. At baseline, most participants were somewhat physically active (i.e., performed ~60 min of moderate physical activity per week) and possessed moderately low to moderately high levels of physical function, based on age- and gender-matched normative data for the Senior Fitness Test (*Rikli & Jones, 1999b*). Program attendance was very high with the
**Table 1** Baseline demographic data for the sample.

|  | Female ($n = 40$) | Males ($n = 18$) | Overall ($n = 58$) |
|---|---|---|---|
| Age (yrs) | $69.7 \pm 5.3$ | $70.0 \pm 5.9$ | $69.8 \pm 4.9$ |
| Height (cm) | $161.9 \pm 5.0$ | $175.9 \pm 6.7$ | $166.3 \pm 8.6$ |
| Mass (kg) | $69.7 \pm 5.3$ | $88.1 \pm 21.0$ | $76.4 \pm 18.4$ |
| BMI (kg m$^{-2}$) | $26.9 \pm 4.8$ | $28.3 \pm 5.2$ | $27.3 \pm 4.9$ |

**Table 2** Frequency statistics for each genotype for all participants ($n = 58$).

| Polymorphism | Genotype | Frequency | Hardy Weinberg equilibrium ($p$-value) |
|---|---|---|---|
| ACE intron 16 | DD | 0.295 | 1.00 |
| Insertion/deletion | ID | 0.492 | |
|  | II | 0.213 | |
| UCP2 G-866A | GG | 0.344 | 0.990 |
|  | GA | 0.541 | |
|  | AA | 0.115 | |

participants attending $22 \pm 2$ out of the requested 24 training sessions, resulting in an overall attendance rate of 93%.

DNA samples were successfully obtained from all participants who commenced the study. However, a small number of subjects did not complete all physical tests during the pre- and post-training assessments. Therefore, the actual sample size for each test differed somewhat for each of the tests, with this presented within the Tables. Importantly, the resulting genotype frequencies for each of the studied polymorphisms conformed to the Hardy-Weinberg equilibrium (Table 2).

## Characteristics and baseline phenotypes

Overall, the frequencies of the *ACE* genotypes were 21, 49 and 30% for II, ID and DD. The distribution of *UCP*2 G-8661 genotypes was 34, 54 and 12% for GG, GA and AA respectively. There were no differences in age, gender, or baseline BMI between genotype groups. As such, all data reported in the tables is for the overall group of 58 male and female participants. However, significant differences were observed for some baseline measures of physical function, when adjusted for age and gender (Tables 3 and 4), specifically between some *ACE* genotype groups for the 6 min walk, 8 ft Up & Go and handgrip strength (Table 3) and *UCP2* genotype groups for 8ft Up & Go and handgrip strength (Table 4).

## Intra- and inter-genotype response to physical training response

Forty-three percent of the overall sample completed all 24 training sessions. Mean attendance of the programme was $22 \pm 2$ out of 24 sessions (93%). No significant difference in attendance rates were observed between male and female participants (data not shown).

**Table 3 Baseline (pre-training) physical abilities according to ACE, adjusted for age and gender.** In the final two columns, the upper row for each assessment provides the p value and in the lower row the value in parentheses is the effect size. Higher scores are indicative of better performance in all tests with the exception of the 8 ft Up and Go in which a smaller score indicates better performance.

| Assessment | ACE II[a] | | ACE ID | | ACE DD | | ACE ID/DD | | IIvIDvDD | IIvID/DD |
|---|---|---|---|---|---|---|---|---|---|---|
| | n | M ± SD | n | M ± SD | n | M ± SD | n | M ± SD | p-value (effect size) | p-value (effect size) |
| 6 min walk (m) | 7 | 458 ± 28.7[*] | 26 | 531 ± 14.6 | 13 | 576 ± 20.7 | 39 | 546 ± 12.1[*] | 0.339 (0.659) | 0.008 (0.614) |
| 8ft Up & Go (s) | 7 | 6.45 ± 0.48[*] | 26 | 4.42 ± 0.24 | 13 | 4.39 ± 0.34 | 39 | 4.41 ± 0.19[*] | 0.219 (0.779) | <0.001 (0.781) |
| Sit to Stand (repetitions) | 7 | 12.4 ± 2.23 | 26 | 13.8 ± 1.14 | 13 | 17.9 ± 1.61 | 39 | 15.2 ± 0.95 | 0.269 (0.726) | 0.268 (0.565) |
| Bicep curls (repetitions) | 7 | 15.4 ± 1.41 | 26 | 16.2 ± 0.72 | 13 | 17.8 ± 1.01 | 39 | 16.7 ± 0.61 | 0.726 (0.273) | 0.421 (0.684) |
| Handgrip strength (kg) | 8 | 30.9 ± 3.01 | 29 | 31.7 ± 1.48[*] | 14 | 29.3 ± 2.18[*] | 43 | 30.9 ± 1.21 | <0.001 (0.786) | 0.994 (0.001) |
| Purdue Pegboard (pegs) | 8 | 11.4 ± 2.01 | 29 | 14.9 ± 0.99 | 13 | 11.7 ± 1.62 | 42 | 14.0 ± 0.85 | 0.451 (0.547) | 0.239 (0.375) |

**Notes.**
[*] Indicates a significant difference between genotype groups for each specific assessment.
[a] Referent group.

**Table 4 Baseline (pre-training) physical abilities stratified by UCP2 G-886A polymporhism, adjusted for age and gender.** In the final column, the upper row for each assessment provides the p value and in the lower row the value in parentheses is the effect size. Higher scores are indicative of better performance in all tests with the exception of the 8ft Up and Go in which a smaller score indicates better performance.

| Assessment | UCP2 G-886A AA | | UCP2 G-886A GA | | UCP2 G-886A GG[a] | | AAvGAvGG |
|---|---|---|---|---|---|---|---|
| | n | M ± SD | n | M ± SD | n | M ± SD | p-value (effect size) |
| 6 min walk (s) | 8 | 523 ± 44.5 | 25 | 539 ± 16.6 | 20 | 516 ± 21.9 | 0.695 (0.336) |
| 8ft Up & Go (s) | 8 | 3.89 ± 0.71* | 25 | 4.47 ± 0.26 | 20 | 5.45 ± 0.35* | 0.045 (0.578) |
| Sit to Stand (repetitions) | 8 | 15.0 ± 3.26 | 25 | 14.5 ± 1.21 | 20 | 14.6 ± 1.60 | 0.992 (0.020) |
| Bicep curls (repetitions) | 8 | 17.5 ± 2.02 | 25 | 15.5 ± 0.75 | 20 | 17.9 ± 0.99 | 0.155 (0.946) |
| Handgrip strength (kg) | 7 | 39.3 ± 3.98 | 29 | 29.9 ± 1.38 | 21 | 30.7 ± 1.95 | 0.094 (0.101) |
| Purdue Pegboard (pegs) | 7 | 13.2 ± 3.07 | 28 | 13.4 ± 1.11 | 21 | 13.8 ± 1.49 | 0.969 (0.031) |

**Notes.**
[*] Indicates a significant difference between genotype groups for each specific assessment.
[a] Referent group.

Tables 5 and 6 show the absolute changes for each physical measure after 12 weeks of training for each genotype group. A trend towards improvement in muscle strength (bicep curls) was observed between *ACE* II v. ID/DD ($p = 0.099$). There was a significant difference between *UCP*2 G-886A genotype groups for the 8ft Up and Go test ($p = 0.023$). Bonferroni correction for multiple testing suggested these differences may be due to chance, as $P > 0.05$ after correction.

## DISCUSSION

The current study's primary aims were to determine if baseline differences and training-related changes in functional performance were related to particular genotypes of the *ACE* and *UCP2* genes in older adults. The frequencies of the *ACE* and *UCP2* G-8661 genotypes in our sample conformed to the expectations of the Hardy-Weinberg equilibrium and to that of a New Zealand population of Caucasian ancestry (*Palmer et al., 2009*; *Palmer et al., 2003*).

**Table 5 Changes in physical measures for each ACE allele after 12 weeks physical training intervention on older adults.** In the final two columns, the upper row for each assessment provides the *p* value and in the lower row the value in parentheses is the effect size. Positive scores are indicative of improvements in performance in all tests with the exception of the 8ft Up and Go in which a negative change indicates better performance.

| Assessment | ACE II[a] | | ACE ID | | ACE DD | | ID/DD | | IIvIDvDD | IIvID/DD |
|---|---|---|---|---|---|---|---|---|---|---|
| | *n* | M ± SD | *n* | M ± SD | *n* | M ± SD | *n* | M ± SD | *p*-value (effect size) | *p*-value (effect size) |
| 6 min walk (m) | 6 | 28.8 ± 14.5 | 26 | 44.2 ± 7.23 | 13 | 44.5 ± 10.0 | 39 | 44.3 ± 5.95 | 0.607 (0.533) | 0.315 (0.703) |
| 8ft Up & Go (s) | 6 | −0.55 ± 0.30 | 26 | −0.23 ± 0.15 | 13 | −0.32 ± 0.20 | 39 | −0.26 ± 0.12 | 0.621 (0.878) | 0.365 (0.916) |
| Sit to Stand (repetitions) | 6 | 1.62 ± 1.58 | 26 | 3.43 ± 0.79 | 13 | 1.25 ± 1.09 | 39 | 2.71 ± 0.67 | 0.217 (0.809) | 0.536 (0.525) |
| Bicep curls (repetitions) | 6 | −0.02 ± 1.14 | 26 | 1.77 ± 0.57 | 13 | 2.50 ± 0.79 | 39 | 2.01 ± 0.47 | 0.198 (0.416) | 0.099 (0.923) |
| Handgrip strength (kg) | 7 | 1.64 ± 1.56 | 29 | 0.78 ± 0.79 | 13 | 2.50 ± 1.15 | 42 | 1.31 ± 0.68 | 0.450 (0.043) | 0.853 (0.054) |
| Purdue Pegboard (pegs) | 7 | −0.31 ± 0.83 | 29 | 0.62 ± 0.42 | 13 | 1.41 ± 0.63 | 42 | 0.85 ± 0.37 | 0.234 (0.965) | 0.189 (0.769) |

Notes.
[a] Referent group.

**Table 6 Changes in physical measures for each UCP2 G-886A allele after 12 weeks physical training intervention in older adults.** In the final column, the upper row for each assessment provides the *p* value and in the lower row the value in parentheses is the effect size. Positive scores are indicative of improvements in performance in all tests with the exception of the 8ft Up and Go in which a negative change indicates better performance.

| Assessment | UCP2 G-886A AA | | UCP2 G-886A GA | | UCP2 G-886A GG[a] | | AAvGAvGG |
|---|---|---|---|---|---|---|---|
| | *n* | M ± SD | *n* | M ± SD | *n* | M ± SD | *p*-value (effect size) |
| 6 min walk (m) | 5 | 39.1 ± 16.9 | 23 | 41.4 ± 7.52 | 17 | 44.3 ± 9.15 | 0.949 (0.397) |
| 8ft Up & Go (s) | 5 | 0.05 ± 0.31* | 23 | −0.10 ± 0.14 | 17 | −0.68 ± 0.17* | 0.023 (0.714) |
| Sit to Stand (repetitions) | 5 | 2.93 ± 1.89 | 23 | 2.85 ± 0.84 | 17 | 2.01 ± 1.02 | 0.789 (0.447) |
| Bicep curls (repetitions) | 5 | 1.00 ± 1.36 | 23 | 1.95 ± 0.61 | 17 | 1.60 ± 0.74 | 0.799 (0.200) |
| Handgrip strength (kg) | 5 | 0.72 ± 1.94 | 26 | 1.07 ± 0.82 | 18 | 2.02 ± 1.03 | 0.708 (0.447) |
| Purdue Pegboard (pegs) | 5 | 1.50 ± 1.05 | 26 | 0.72 ± 0.45 | 18 | 0.38 ± 0.56 | 0.622 (0.319) |

Notes.
[a] Referent group.

At baseline, it was observed that the *ACE* gene variant genotypes were associated with significant differences in 3 of the 6 performance tests (6 min walk, 8ft Up and Go and handgrip strength). Baseline variations in the *UCP2* gene polymorphism were also observed for the 8ft Up and Go, with handgrip strength approaching significance ($p = 0.094$).

The ID/DD *ACE* allele groups performed significantly better at baseline than those with those with II alleles on the 6 min walk and 8ft Up and Go tests. While these two walking tests both require the participants to walk as quickly as possible over a set course, these tests differ in several ways. The 6 min walk test requires the participants to walk a maximum distance in 6 min, and is therefore considered a field test of $VO_{2peak}$ and walking ability for older adults (*Rikli & Jones, 1999a*; *Rikli & Jones, 1999b*). In contrast, the 8ft Up and Go requires the participants to stand up, walk a distance of 8 ft, turnaround, walk back to the chair and then sit down, suggesting a greater reliance on lower body strength, walking speed and mobility (*Rikli & Jones, 1999a*; *Rikli & Jones, 1999b*).

The baseline differences for the 8ft Up and Go appear consistent with previous studies demonstrating increased muscular strength or power with the D allele of the *ACE* gene for younger adults (*Puthucheary et al., 2011*; *Woods et al., 2001*), but in contrast to some studies involving older adults (*Garatachea et al., 2012*; *Giaccaglia et al., 2008*; *Pereira et al., 2013*). While the greater 6 min walk distance for the ID/DD groups was initially unexpected, a lack of lower body strength may reduce the ability of older adults to walk briskly for extended periods of time. Research support for this view is provided by the significant improvements in $VO_{2peak}$ observed in older adults as a consequence of resistance training (*Lovell, Cuneo & Gass, 2009*). The significantly greater performance in the two walking tests for the ID/DD groups is also consistent with a recent report (*Seripa et al., 2011*), observing that the *ACE* II genotype was associated with greater limitations in activities of daily living in hospitalised older patients. As gait speed over a variety of distances is an influential determinant of maintaining independence and in reducing many age-related adverse effects (*Abellan van Kan et al., 2009*), the results of our study and (*Seripa et al., 2011*) suggest some genetic component to the wide variation seen in walking speed and performance of older adults (*Peel, Kuys & Klein, 2013*).

Based on the reported role of the *ACE* gene in human physical performance studies involving younger adults (*Dhamrait et al., 2012*; *Puthucheary et al., 2011*), our findings at baseline for the 30 s STS and handgrip strength test were somewhat unexpected, with greater performance expected from those with the ID/DD alleles for both of these tests. A significant between-group was observed for handgrip strength, whereby the ID group had greater handgrip strength than the DD group. While such a result was not expected based on the results of the younger adult literature, it was consistent with the only other study conducted to date involving older adults (*Bustamante-Ara et al., 2010*). Our results indicated no significant baseline differences between *ACE* genotype groups for the 30 s STS. While this may have unexpected based on the younger adult literature, the lack of significant baseline differences in 30 s STS performance was also consistent with the results reported for 139 older women by *Pereira et al. (2013)*.

Comparisons of the baseline performance results for the *UCP2* genotype groups also revealed significant differences for the 8ft Up and Go test and a trend for handgrip strength ($p = 0.094$), whereby greater performance was found in those with the A than G *UCP2* allele. This result was somewhat consistent with (*Dato et al., 2014*), who reported significantly greater walking speeds over 3–4 m in older adults with the *UCP2* A allele. However, the direction of these baseline differences for the 8ft Up and Go test was inconsistent with the greater improvements in endurance performance markers for the A compared to G allele carriers in younger populations (*Dhamrait et al., 2012*; *Perusse et al., 2013*). Such inconsistencies in the association between *UCP2* polymorphisms and endurance vs power performance in younger compared to older adults warrants further investigation.

The second aim of the current study was to examine whether the *ACE* and *UCP2* gene influenced the training related responses in older adults' functional performance. The training program used in the current study appeared effective as the magnitude of the training related changes for the Functional Fitness Test battery outcomes was relatively

consistent with that reported for other community-based older adult exercise programs (*Bates et al., 2009*; *Belza et al., 2010*; *Henwood, Wooding & de Souza, 2013*). Inspection of the data revealed little evidence for the *ACE* or *UCP2* gene alleles to have any significant effects on the training-related changes in the outcome measures assessed in this study. Exceptions to this rule were the significantly greater improvements for the GG *UCP2* group in the 8ft Up and Go and the trend ($p = 0.099$) for larger improvements in bicep curl strength for the ID/DD versus II *ACE* allele groups. These associations between the *UCP2* gene and training-related changes in the 8ft Up and Go test are novel and to the authors' knowledge, our study is the first to assess the relationship between the *UCP2* genotypes and changes in physical performance in older adults.

The lack of differences in the training related response between different *ACE* genotype groups was consistent with the only study to assess changes in older adults' 6 min walk performance resulting from 18 months of combined aerobic and resistance training (*Giaccaglia et al., 2008*). The responses in our assessments that were more dependent on strength or power were inconsistent with the two studies conducted to date, where significantly greater improvements in muscular strength and power were associated with the *ACE* D gene allele (*Giaccaglia et al., 2008*; *Pereira et al., 2013*). Potential reasons for the variation in our results compared to the two studies within the literature may be explained by three primary factors.

The first of these was the difference in outcome measures assessed in the current study compared to that of the literature. The current study focused on functional performance tests (30 s STS, bicep curl, 8 ft Up and Go and 6 min walk), whereas most of the outcomes assessed in the literature were strength or power tests. While performance in many of the functional performance tests we selected is moderately correlated with strength and/or $VO_{2peak}$, our assessments were not criterion measures of either of these physical qualities. It is therefore likely that functional performance tests have a greater variety of determinants than criterion tests of muscular strength or $VO_{2peak}$, thereby reducing the potential for isolated genotypes to significantly influence training-related changes in functional performance. Such speculation is consistent with the lack of any significant relationship between the *ACE* and *UCP*2 genotypes for the older adults' baseline- or training-related differences in upper limb dexterity (Purdue Pegboard) test performance seen in this study. The second difference may have been the relatively high levels of baseline function for some genotype groups in the current study compared to that of the literature. Specifically, the *ACE* ID/DD and *UCP2* AA gene allele groups who had the highest scores at baseline, were typically in the 60–80th percentile (men) and 80–90th percentiles (women) for the 30 s STS, bicep curl, 6 min walk and 8 ft Up and Go tests (*Rikli & Jones, 1999b*). In comparison, our other gene allele group and older adults in the other studies within this literature appeared to have much poorer baseline function. These weaker older adults in ours and other studies therefore had more chance to demonstrate significant improvements in function and significant genotype-related responses to training than our higher functioning ID/DD and *UCP*2 AA groups. Thirdly, the current study had more modest statistical power for detecting training-related changes (70% power) than baseline differences (90% power).

In conclusion, the current study adds to the literature on the potential link between genotype and physical performance in older adults. While our baseline results were generally consistent with the phenotypes associated with the *ACE* I/D and *UCP2* polymorphisms, we found no strong evidence of genetic association to the functional performance response to physical training in this study. Some of this relative equivalence may have reflected our focus on functional performance such as the 8ft Up and Go test rather than physical capacity outcomes e.g., MVC strength or aerobic power. Specifically, functional performance assessments would appear to require the integration of a greater number of sensoriomotor functions than physical capacity assessments of MVC or $VO_{2peak}$. The variance in results for this study compared to that of literature may also reflect the relatively high levels of baseline function for some genotype groups of the older adults in the current study as well as the greater statistical power for detecting baseline differences (90% power) than the training-related changes (70% power). Future studies in this area should recruit larger sample sizes so to obtain sufficient statistical power to detect meaningful training related responses in a variety of health, physical fitness and functional performance outcomes and ensure no statistically significant between-group differences in outcome scores at baseline. These future studies should also be sufficiently powered to detect differences in training response as a function of interactions between different polymorphisms of multiple genes that may play a role in modulating the exercise response.

### Funding

The Auckland University of Technology's Faculty of Health and Environmental Sciences funded this project through the CGH 10/06 AUT Contestable Grant round. The funders had no role in study design, data collection and analysis, decision to publish, or preparation of the manuscript.

### Grant Disclosures

The following grant information was disclosed by the authors:
Auckland University of Technology's Faculty of Health and Environmental Sciences.

### Competing Interests

The authors declare there are no competing interests.

### Author Contributions

- Justin W.L. Keogh conceived and designed the experiments, performed the experiments, contributed reagents/materials/analysis tools, wrote the paper, prepared figures and/or tables, reviewed drafts of the paper.
- Barry R. Palmer conceived and designed the experiments, performed the experiments, analyzed the data, contributed reagents/materials/analysis tools, wrote the paper, prepared figures and/or tables, reviewed drafts of the paper.

- Denise Taylor conceived and designed the experiments, contributed reagents/materials/analysis tools, prepared figures and/or tables, reviewed drafts of the paper.
- Andrew E. Kilding conceived and designed the experiments, analyzed the data, contributed reagents/materials/analysis tools, prepared figures and/or tables, reviewed drafts of the paper.

## Human Ethics

The following information was supplied relating to ethical approvals (i.e., approving body and any reference numbers):

Ethical approval for the studies was given by the Auckland University of Technology Human Ethics Committee. The ethics approval number was 06/05.

## Supplemental Information

Supplemental information for this article can be found online at http://dx.doi.org/10.7717/peerj.980#supplemental-information.

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
