# Peer review of "ACE and UCP2 gene polymorphisms and their association with baseline and exercise-related changes in the functional performance of older adults"

_PeerJ, doi:10.7717/peerj.980_

## Round 0.1 · original submission · Major Revisions

· Academic Editor

Major Revisions

Please prepare a point-by-point response that states how and where each issue raised by reviewers and the editor (as listed below) have been addressed in the revised manuscript. In addition to the issues raised by reviewers, I have the following concerns:

- more information on the N2O trial including recruitment rates, adherence rates and other baseline demographic, body composition and other covariates should be provided in the Methods and Table 1. Please clarify if this was a single arm trial.

-more information on the genotyping methods including QA/QC measures should be provided.

-please provide more information on the statistical methods including how HWE was evaluated, how potential population stratification was addressed, the plan for multiple testing correction, how baseline covariates were evaluated for potential inclusion in the model(S). Clarify if additive and/or dominant and recessive genetic models were all evaluated. Please clarify the issue of statistical power (the methods state sufficient power to detect effects but the discussion indicates insufficient sample size).

- the abstract and results (and Tables) should present effect sizes as well as p-values for the specific genotype(s) where an association was observed. Furthermore, clarify the referent group in these sections as well as presentation in the Tables. Discuss (and present in Tables if relevant) any differences in allele frequency by gender.

·

Basic reporting

In this article, the authors was to determine if ACE and / or UCP2 genotypes: 1) Distinguish the baseline level of functional performance in older adults who are not performing an exercise program Currently; and 2) influence the training-related changes in older adults' functional performance. The main conclusion of this study is that the authors not found strong evidence of genetic association to the functional response performance to physical training in this study.

Abstract: The abstract is informative and explanatory self and includes the significant results that support the completion of the work.Introduction: The introduction is comprehensible to general reader. Clearly puts the work proposal and provides relevant evidence supporting the importance and meaning of work. But in the introduction, in line 46 first appears MVC acronym suggest describe.

Experimental design

The research is relevant and meaningful mainly to professionals who prescribe exercises. It is emphasized that the investigation was conducted with scientific rigor fact that observed in the method description. Provides sufficient information in the text so that the method used can be repeated in other studies without the need for communication with the authors.The research was conducted in conformity with the prevailing ethical standards in the field.

Validity of the findings

The data date are robust, statistically sound, and controlled. The statistical method was used correctly. However, I for data arranged in the table suggest enter a differentiation (such as an asterisk *) in groups when there is statistical difference to further elucidate the presentation of data and to facilitate the reader. In the discussion the authors could emphasize more clearly the aspects most important to study . The conclusions are Appropriately stated and are connected to the original question investigated, and is based by the results presented.

Additional comments

The theme is relevant to the academic community especially to support exercise prescription for the elderly safely and effectively considering the particularities regarding genetic predictors of physical function.

Reviewer 2 ·

Basic reporting

The manuscript entitled “ACE and UCP2 gene polymorphisms and their association with baseline and exercise-related changes in the functional performance of older adults“ is an original article on the associations between ACE I/D and UCP2 -866 G>A polymorphisms and physical performance in older adults.
Relevant prior literature appropriately referenced.
Methods described with sufficient information to be reproducible by another investigator.
The research conducted in conformity with the prevailing ethical standards in the field.
Quality of written English: The article is written in English using clear and unambiguous text and conforms to professional standards of courtesy and expression. Some language corrections need to be done before being published.
Level of interest: An article which findings are important to those with closely related research interests.

Experimental design

The experiment presented in the paper is well planned. However, the Reviewer considers that there are some issues that should be addressed.

Validity of the findings

No Comments

Additional comments

1. There is a lack of information about the function of ACE and UCP2 genes. The authors have not offered scientific rationale for interaction between the ACE and UCP2 genotypes. Given that this is the premise of the research, this should be included.
2. More comprehensive review of literature is required for the genetic studies of older adults. For example, the latest papers Dato S., et al. Contribution of genetic polymorphisms on functional status at very old age: A gene-based analysis of 38 genes (311 SNPs) in the oxidative stress pathway (2014); Hai Bo., et al. Mitochondrial redox metabolism in aging: Effect of exercise interventions (2013).Genes, mutations, genotypes, and alleles should be indicated in italics, and authors are required to use approved gene symbols, names and formatting. In that case, gene names should be rewritten in Italic.
3. Introduction, line 74: add names for polymorphisms ACE I/D and UCP2 G-866A
4. In the methods, please provide more information associated with the distribution of sex and age (range or mean (SD)) among participants.
5. Methods, line 146: please add to the description of isometric handgrip strength, if the handgrip strength was measured for right or left hand
6. Result, line 189 in "BMI" should not be included. If there is a wish to add BMI description, it should be done in the methods section.
7. There should be a description of MVC strength and VO2 peak.

To sum up, despite the fact that the number of participants was relatively small, the authors have produced a lot of interesting data which deserves publication.

---

## Round 0.2 · Minor Revisions

· Academic Editor

Minor Revisions

There are still some concerns that reviewer 3 noted that need addressed. Please address these and provide a point by point response on how and where each issue was addressed.

Reviewer 2 ·

Basic reporting

No Comments

Experimental design

No Comments

Validity of the findings

No Comments

Additional comments

The corrections are very well made and overall the authors have produced a lot of interesting data which deserves publication.

·

Basic reporting

"No comments".

Experimental design

"No comments".

Validity of the findings

In the session results, it was not possible to observe the results of the PAR-Q test, it was reported that almost all participants had at least one co-Morbidad but was not demosntrada morbidity which it was written that the participants were little physically active, but not got to know how weak they were.

How many people were not mentioned started the study, neither the exclusion criteria of the study, I was wondering how many days during the training period people could be missing (not go to training).

I could not see which were the exercises and their intencidades, I wonder if there were any maximum strength test (The 1RM testing for prediction) for the prescription of exercises.

I believe that one of the limitations that should appear in the text, is not the measurement of fat percientual as this reported in the literature that the BMI is not a good parameter to check body fat old, this factor may be affected the results.

Additional comments

An interesting research, but with some methodological flaws that do not allow the same be reproduced without contact RESEARCHERS.

There are limitations that should be better descirtas in the text, in order to provide greater transparency on the results.

---

## Round 0.3 · accepted · Accept

· Academic Editor

Accept

All issues have now been fully addressed to satisfaction of reviewers.

·

Basic reporting

No Comments

Experimental design

No Comments

Validity of the findings

No Comments

Additional comments

No Comments